# Reproductive Capacity and Scanning Electron Microscopy (SEM) Analyses of the Micromorphological Surfaces of Three Endemic *Satureja* Species from Bulgaria

**DOI:** 10.3390/plants12132436

**Published:** 2023-06-24

**Authors:** Ivanka Semerdjieva, Elina Yankova-Tsvetkova, Valtcho D. Zheljazkov, Lyubka H. Koleva-Valkova, Rozalia Nikolova

**Affiliations:** 1Department of Botany and Agrometeorology, Agricultural University, Mendeleev 12, 4000 Plovdiv, Bulgaria; v_semerdjieva@abv.bg; 2Department of Plant and Fungal Diversity and Resources, Institute of Biodiversity and Ecosystem Research, Bulgarian Academy of Sciences, 1113 Sofia, Bulgaria; rozalianikolova23@gmail.com; 3Department of Crop and Soil Science, Oregon State University, 3050 SW Campus Way, 109 Crop Science Building, Corvallis, OR 97331, USA; valtcho.jeliazkov@oregonstate.edu; 4Department of Plant Physiology, Biochemistry and Genetics, Agricultural University, Mendeleev 12, 4000 Plovdiv, Bulgaria; l_koleva2001@yahoo.com

**Keywords:** pollen, embryology, nutlets, leaves, surfaces, germinations, male and female gametophyte, embryo, endosperm

## Abstract

*Satureja pilosa* Velen., *S. coerulea* Janka and *S. kitaibelii* Wierzb. ex Heuff. are Balkan endemic species, and they are distributed in restricted territories, primarily found in dry grasslands, rocky slopes, and stony habitats. This study presents the results of the first embryological and micromorphological analyses of three *Satureja* species (*S. pilosa*, *S. kitaibelii*, and *S. coerulea*) from the Bulgarian flora. The aim of this study was to establish the features of the male and female reproductive sphere, as well as surface characteristics of leaves, stem, and calyx in order to understand the mode of reproduction, character, size and state of species populations and delimitation. For the embryological study, flowers and flower buds in different developmental stages were collected from plants of natural populations and treated with the classic paraffin method. Reproductive capacity was assessed using the following approaches: (1) acetocarmine test for pollen viability, (2) tetrazolium test (TTZ) for seed viability, and (3) germination test. The surfaces characteristics of leaves, stem, and calyx were analyzed by scanning electron microscopy (SEM). As a result, the study revealed the structures of the male (M) and female (F) generative spheres as well as the processes of gamete development, pollination, and endosperm and embryo formation. It was found that the three *Satureja* species exhibited a high pollen viability but low seed viability and germination. The SEM analysis showed both common and distinct micromorphology features regarding epidermis, calyx and stem surfaces among the three *Satureja* species. Notably, the *S. coerulea* surfaces (leaves, calyx, stem) were clearly distinguishable from the other two *Satureja* species. Regarding the nutlet surfaces, it was observed that the exocarp surfaces of *S. coerulea* and *S. kitaibelii* have a reticulate convex type surface and comprise two types of cells: (1) highly bulging, bubble-like cells; and (2) flat cells with numerous striations. On the other hand, the exocarp of *S. pilosa* displayed rectangular or polygonal shapes without bubble-like cells, and it had a tabular to slightly convex type surface. Additionally, nutlets (seeds) of both *S. coerulea* and *S. pilosa* exhibited distinct papilla formations resembling non-glandular trichomes seen on the ends of the nutlets for *S. coerulea* and over the entire surface for *S. pilosa*. The characteristics of the embryological structures and processes, along with the absence of apomixis, characterized the three studied *Satureja* species as sexually reproducing. The established balanced processes and stable structures contribute to their high reproductive potential and population stability. However, these traits may also decrease their adaptability to environmental changes.

## 1. Introduction

*Satureja* is one of the largest genera in the Lamiaceae family [1]. The different *Satureja* plants are commonly often used as culinary herbs, and they are widely cultivated [2,3]. Most *Satureja* species have been studied for their chemical composition and biological activities [4,5,6,7]. The species of this genus are distributed in various regions of the world, and most of the species are found in the Mediterranean region and surrounding areas [8]. According to Flora Europaea, there are 12 naturally distributed *Satureja* species across Europe, and five of them are found in Bulgaria [9].

*Satureja pilosa*, *S. coerulea* and *S. kitaibelii* are Balkan endemic species. They are naturally distributed in the Bulgarian flora in dry grasslands, rocky slopes, and stony habitats under extreme conditions [6,7,9]. The *Satureja* species have multiple mechanisms to adapt and survive in these unfavorable environmental conditions. These mechanisms include dry tolerance (small leaves with a thick, waxy cuticle), cold tolerance (high-altitude habitats), allelopathy (through the production of allelopathic compounds that inhibit the growth of competing plant species) [10], chemical defense (through the production of essential oil (EO) containing antimicrobial, antifungal, and insecticidal compounds) [6,7], and reproductive strategies. Due to the increased interest in these species and their limited distribution, it is important to implement measures to protect their gene pool and ensure their long-term survival. Generally, endemic species are only found in a restricted geographic location and are often of high conservation significance [11]. Ex situ, in situ collections and biotechnological tools are important approaches for conservation of endemic plants for several reasons: (1) they can help to preserve plant diversity by maintaining populations; (2) they are a source of plant material for research; (3) help with the development of new crops; (4) preserve the genetic diversity of endemic species, etc. [11]. Reproduction plays a key role in plant species and population biology, as well as in the perpetuation of the species [12,13]. The successful cultivation and preservation of rare and endemic plant species requires comprehensive information on their reproductive strategy [14,15,16]. General knowledge on the reproductive strategy of threatened and endemic plants plays a key role not only in systematic studies and evolution [17] but also in the development of effective conservation strategies [18].

The review of the literature sources showed limited data on the embryological structures and processes, type of reproduction and reproductive capacity of species in the genus *Satureja.* Only the flower structure (pollination system and self-incompatibility) in *S. sahendica* Bornm. and *S. bachtiarica* Bunge have been reported [19]. On the other hand, a number of other species in the Lamiaceae family have been subjects of embryological studies [20,21,22,23]. The embryological features of *Nepeta cataria* L., *Salvia nemorosa* L., *Lamium maculatum* L., *L. garganicum* L., and *L. bifidum* Cirillo are well-documented [20,21,22,24]. These studies reported that anthers were tetrasporangiate and four-layered; tapetum was of the secretory type; and meiosis in the anthers was of simultaneous type. Furthermore, the cited authors established three-celled hexacolpate pollen, which is typical for the Lamiaceae species [20,21,22,24]. Overall, the embryological features of plants provide valuable information about the plants’ reproductive biology, including aspects such as pollen development, fertilization, and seed formation.

Scanning electron microscopy (SEM) analyses, on the other hand, provide a detailed view of the surface morphology of various plant parts (leaves, stems, flowers, pollen, seeds). SEM is a powerful tool for describing the surface morphology, microstructure and subcellular ultrastructure of plants [25]. This information can be useful in identifying different species of *Satureja* plants, as well as understanding their adaptations to different environmental conditions. Several papers reported SEM analyses of *Satureja* species, mainly for *S. hortensis* L., and *S. montana* L. [26,27,28,29]. Indeed, these *Satureja* species are widely cultivated for their EO, which is synthesized and accumulated in glandular trichomes [6,30]. Furthermore, SEM analyses could provide useful information in studying the surfaces and morphology of leaves, nutlets (seeds) and pollen, especially for *Satureja* species where there are several unresolved taxonomic questions [31]. These issues were a result of difficulties in species delimitation for the following reasons: (1) *Satureja* species can easily hybridize within the same population as well between different *Satureja* species [32]; (2) subsequently, there is a high degree of morphological variability and overlapping characteristics within and among different species because of hybridization [31,32,33]; (3) there are geographical variations in morphology [32]. Comparative SEM analyses of the surface features of nutlets (seeds), leaves, and pollen in the three Balkan endemic species *S. pilosa*, *S. coerulea* and *S. kitaibelii* distributed in the Bulgarian flora have not been reported. The objectives of this study were (1) to establish the features of male and female reproductive spheres of *S. pilosa*, *S. coerulea* and *S. kitaibelii* in connection with revealing the mode of reproduction, character, size and state of their populations; and (2) to analyze and compare surfaces of nutlets (seeds), leaves, calyx and stem in order to provide new data for species knowledge and delimitation.

## 2. Results

### 2.1. Embryological Analyses

The embryological characteristics observed in the study were consistent with those described for the *Lamiaceae*, and were found to be largely similar across the studied species. Therefore, the images and description of results from this study are common for *S. pilosa*, *S. coerulea* and *S. kitaibelii.*

#### 2.1.1. Anther and Development of the Male Gametophyte

The anther was four-locular (Figure 1A). The anther wall was four-layered, arranged according to the dicotyledonous type of the Davis classification [34]. The four-layered walls consisted of epidermis, endothecium, middle layer and secretory tapetum. The epidermis was represented by single row of one-nucleate cells. Their forms were rectangular, and during anther ontogenesis, they enlarged tangentially and rounded up outside. The middle layer was one-rowed. The shapes and sizes of cells of the middle layer were heterogeneous. This layer degenerated about the homotypic division of meiosis in microspore mother cells (MMCs). At the stage of one-nucleate pollen (Figure 1D), the endothecium developed fibrous thickenings. Initially, the tapetum was glandular with one-nucleate cells. During anther ontogenesis, highly elongated tapetum cells underwent rapid multiplication of nuclei due to successive mitotic divisions without subsequent cytokinesis. After the formation of unicellular pollen, the tapetum cells enlarged and intruded into the anther locules and formed “placentoids” (Figure 1D). This layer remained cellular until the maturity of the anther. The sporogenous tissue was one-rowed (Figure 1B). The primary sporogenous cells functioned directly as MMCs. After meiosis and simultaneous microsporogenesis consecutively running in them predominantly, tetrahedral tetrads formed (Figure 1C). The mature pollen grains were two-celled (Figure 1E).

#### 2.1.2. Ovule and Development of the Female Gametophyte

The ovary in the studied species was syncarpous, superior, and four locular (Figure 2A). Each locule consisted of a single anatropous ovule, with tenuinucellate unitegument on axile placentation (Figure 2B,C). In the early stages of ovule development, a unicellular archesporium formed below the surface, directly functioning as the microspore mother cell. (MMC). The archesporogenesis proceeded without cover cell cutting. As a result of macrosporogenesis in MMC, a linear macrospore tetrad was formed (Figure 2D). We observed that the embryo sac started its development of the micropylar megaspore, besides the chalazal megaspore, and one embryo sac in the ovule locule was observed.

The female gametophyte in the studied *Satureja* species followed the basal *Polygonum* type of development (Figure 3A). The mature embryo sac (ES) is typical for this type of structure and arrangement of its elements. The arrangement of elements was as follows: (1) egg apparatus composed from an egg cell and two hook synergids with filiform apparatus at the micropylar end of ES; (2) two polar nuclei situated near the egg apparatus, below the egg cell; and (3) three antipodals at the chalazal end of the ES (Figure 3A). The embryo and endosperm started their development after double porogamous fertilization. The endosperm was cellular but initially passed through a free nuclear stage. Embryogenesis began after endosperm genesis and can be defined as the *Onagrad type* according to Johansen’s classification of embryogenesis [35,36]. Apomixis was not observed. Proterandry was found to take place in flower development: when mature pollen is formed in the anthers, a tetrad megaspore is formed in the ovule (Figure 3C).

#### 2.1.3. Nutlets (Seed) and Pollen Viability Testing

According to the requirements of tetrazolium methods (TTZ test) [37,38], the seeds of *S. coerulea, S. kitaibelii* and *S. pilosa* were divided into six classes (Figure 4). The first class (Class I) included the seeds with embryos stained in dark red (Figure 4A). The second class (Class II) included the seeds with pink colored embryos (Figure 4B). The third class (Class III) included the seeds with embryos in which only the root was stained in red (Figure 4C). The fourth class (Class IV) included the seeds with colorless embryos (Figure 4D). The fifth class (Class V) comprised the empty seeds (Figure 4E). The sixth class (Class VI) included the seeds with undeveloped (dried) embryos (Figure 4F). According to Moore’s classification [39], the seeds in Classes I, II and III were considered as viable, and those in Classes IV, V and VI were considered unviable. Therefore, the amount of viable seeds (embryos) estimated in % was as follows: *S. pilosa*: 31.18%, *S. coerulea*: 39.58%, *S. kitaibelii*: 10.58% (Figure 5).

After acetocarmine staining, the cytoplasm and nuclei of viable pollen grains were stained in red, while unviable, empty and shrunken pollen grains remained unstained (Figure 6). The results of the study showed high viability of the mature pollen: 90.01 ± 3.35 for *S. pilosa*, 94.14 ± 3.82 for *S. coerulea*, and 93.44 ± 3.88% for *S. kitaibelii*.

#### 2.1.4. Nutlet (Seed) Germination

*Satureja kitaibelii* seeds were not tested for germination due to insufficient seeds for three replicates because most of the seeds of this species were empty and non-viable. Overall, the seeds of *S. coerulea* and *S. pilosa* were tested for germination energy (GE) and germination (G). The results are presented in Table 1. Low germination energy and germination rates (below 50%) were observed in both species—*S. coerulea* and *S. pilosa*. Both indicators (GE and G) were the lowest in variant 2 (fluorescent white light with addition of LED red and blue in ratio of 7:1) for *S. pilosa* (9 and 18%). The highest GE values were observed for variant 3 (fluorescent white light with addition of LED red and blue in ratio of 4:1) in *S. coerulea* (31%). Natural light (V 1) was found to be the most suitable for seed germination (G%) compared to applied artificial light combinations in both tested species.

### 2.2. Scanning Electron Microscopy (SEM) Analyses

#### 2.2.1. Leaves, Stem, and Calyx Surfaces

Overall, all studied *Satureja* species parts (leaves, stem, calyx, corolla) were covered with trichomes. The indumentum of this species presented glandular and non-glandular trichomes.

##### *Satureja coerulea* 

The micromorphological characteristics of *S. coerulea* (leaves, calyx, and stem) are presented in Figure 7A–F, Figure 8A–D and Figure 9A–C. The leaves are small and smooth. The SEM analyses showed that the main epidermal cells are irregular to rectangular in shape, with slightly raised to smooth periclinal walls. The stomata are elliptical and submerged in the epidermis, with distinct cuticles and scattered epicuticular wax plates (Figure 7A,E). Both non-glandular and glandular trichomes are present on the leaf surface, with the latter being peltate and located on the abaxial surface (Figure 7A–D). Stereo microscope analyses showed that the calyx is short, with five shallowly incised teeth, ten veins, and anthocyanin coloring (Figure 8). The calyx surfaces have highly undulating epidermis surfaces with convex periclinal walls, and both non-glandular and glandular trichomes are distributed over the entire surface (Figure 8A–D). The stem surface is smooth to raised with striations and covered with unicellular to multicellular, unbranched non-glandular trichomes and epicuticular wax plates (Figure 9A–C).

##### *Satureja kitaibelii* 

The SEM analyses of *S. kitaibelii* revealed that the epidermal surfaces of its leaves are smooth to slightly convex with striation (Figure 7G–I). The main epidermal cells are isodiametric, with a rounded edge and straight anticlinal walls. The waxes are continuous layers with single crusts of varying thicknesses (Figure 7G–I). The stomata are located on both surfaces, and they are mostly diacytic. The cells of the stomatal complex of *S. kitaibelii* have specific striations (Figure 7I,K). The guard cells are round shaped and small. The observed indumentum includes both non-glandular and glandular peltate-type trichomes on both surfaces of leaves (Figure 7G–L). The non-glandular trichomes are of two types: multicellular and unbranched along the edge of leaves (Figure 7J), and unicellular and uniseriate as papilla (Figure 7G,H,L). The stereo microscope analyses showed that the calyx of *S. kitaibelii* is cylindrical, slightly tubular, 6–7 mm long, with 10 veins and anthocyanin coloring (Figure 8). The calyx surfaces are highly wrinkled, with multiple non-glandular and glandular capitate- and peltate-type trichomes (Figure 8E–H). The stem surfaces are covered with waxes and two types of non-glandular trichomes, namely unicellular conical curved and multicellular unbranched (Figure 9D–F).

##### *Satureja pilosa* 

The leaves of *S. pilosa* are oblong-lanceolate, acuminate. SEM analyses of the leaf surfaces showed that the main epidermal cells are isodiametric with curve anticlinal walls (Figure 7). The edges of the anticlinal walls are oval to point, and the periclinal walls are striated (Figure 7N–R). The stomata are oval shapes with a double rim of waxes at the level of the epidermis (Figure 7O,Q,R). The stomata are in both surfaces (ad, ab), and leaves are amphistomatic (Figure 7). Overall, the whole plant is covered with non-glandular simple conical trichomes and glandular peltate- or capitate-type trichomes (Figure 7 and Figure 8). The wax crusts are observed over the entire surface. The sepals are cylindrical and elongated (5–6 mm), and the teeth are triangularly incised and covered with multicellular simple, conical hairs and peltate-type glandular trichomes (Figure 8I–K). The stem is the most hairy, with longitudinal striations (Figure 9G–I). The surface of the epidermal cells of the stem is ribbed, with indented and slightly convex ribs covered with a waxy film. Wax plates are observed over the entire surface of the stem. The non-glandular hairs are unicellular or multicellular (mostly four-cellular), conical, with papillose surfaces. The glandular trichomes are of the capitate type with a short stalk and a large head.

#### 2.2.2. Nutlet (Seed) Surfaces

The seed surfaces and morphology of three *Satureja* species were analyzed with stereo microscope and SEM analyses. The pictures are presented in Figure 10 and Figure 11.

##### *Satureja coerulea* 

The nutlets of this species are dark brown, with lighter speckled areas and indistinct dark stripes that start from the top and reach toward the middle (Figure 10(1A–1D)). The nutlets of *S. coerulea* varied in size (from 113.32 μm to 182.38 μm) and form. The form varied from very small, round, bubble-like to elongated ovals (Figure 10(1A)). On the abaxial side of the nutlets, there is a ridge that starts from the middle of the surface. The tips of the nutlets are uneven and triangularly beveled on both sides (Figure 10(1C)). The exocarp surfaces have specific papilloma formations seen on the end of the nutlets (light microscope, Figure 10(1A–1C)). The SEM analysis showed that the surface of the exocarp is a reticulate convex type and comprises two types of cells. The first type consists of highly bulging, bubble-like cells and flat cells with numerous striations (Figure 11A–D). A thickened rim is observed at the base of the bulging cells. Comb-like connections are established between the two types of cells. The anticlinal walls are straight with slight striations, and the periclinal walls vary from smooth to wavy grooves. The waxes are observed all over the surface.

##### *Satureja kitaibelii* 

The nutlets of *S. kitaibelii* are dark brown, flattened oval, with a winged triangular pointed tip, and length varied from 67.95 μm to 76.38 μm (Figure 10(2A–2D)).

Dark brown stripes are observed on the upper side from the tip, which starts from the top and reaches the middle of the nutlets (Figure 10(2A)). The nutlet of this *Satureja* species has a triangular edge in the middle from its lower part. The micromorphological SEM analyses showed that the epidermal cells of the exocarp are composed of two types. The first type of cell is with rhomboid forms, strongly convex, while the second type of cell is with polygonal shapes, tabular type (Figure 11E–H). The anticlinal walls are undulating with a rounded edge, while the periclinal walls are smooth with striations. According to the terminology and classification described by Barthlott and Ehler (1977), the surface of *S. kitaibelii* is convex type. The entire surface is covered with waxes.

##### *Satureja pilosa* 

The seeds of *S. pilosa* are dark brown, from round to oval elongated forms, and varied in size from 129.22 μm to 140.07 μm. The tips of seeds are triangularly beveled, winged, and in lighter brown colors. On the adaxial surfaces of the seeds, there are three to four different lengths of longitudinal brown stripes that start from the tip, and they are located along the entire length of the seed (Figure 10(3A–3D)). The tip is arrow-shaped, beveled, and laterally, it is triangularly concave, with a lighter color (Figure 10(3A–3D)). Formations like warts (papillae) can be observed on the entire surface of the seeds (Figure 10(3A–3D)). The micromorphological SEM analyses showed that the epidermal cells of exocarp are of rectangular or polygonal shapes (Figure 11I–L). The anticlinal walls are undulating, with a pointed edge and indistinct striations, while periclinal walls are smooth to wavy convex. According to the terminology and classification described by Barthlott and Ehler (1977), the surface of *S. pilosa* is tabular to slightly convex type. The entire surface is covered with waxes.

## 3. Discussion

### 3.1. Embryological Analyses

This is the first study on the sporogenesis and gametogenesis of the three Balkan endemic *Satureja* species. The established features of the structures and development of gametophytes (M,F) are similar to those described in other species from Lamiaceae [20,21,22,23,34,40,41,42,43]. The main characteristics of gametophytes (M,F) of *Satureja* species are: four-sporangiate anthers with four-layered walls, tetrahedral microspore tetrads, anatropous tenuinucellate ovule, development of *Polygonum*-type female gametophyte and *Onagrad*-type embryo. The formation of placentoids, observed at the phase of one-cellular pollen that coincides with the beginning of degeneration of the tapetum, was described earlier in other Lamiaceae species such as *Sideritis scardica* Griseb [44] and *Marrubium friwaldskyanum* Boiss [45]. Furthermore, this structure was reported as a typical feature of the genera *Lavandula*, *Salvia*, and *Stachys* from Lamiaceae [41]. The observation of this structure in other genera such as *Sideritis* and *Marrubium*, as well as in this study, suggests that placentoids are a typical feature of the species belonging to the Lamiaceae family. Most of the reported data for other species from Lamiaceae showed that the chalazal megaspore of the tetrad was the functional megaspore, which becomes the embryo sac mother cell (MMC) [34,40,41,42,43]. In this study, it was established that in the *Satureja* species, the female gametophyte starts its development not only from the chalazal megaspore, but also from the micropylar one. The presence of more than one functional megaspore has been observed in other representatives of Lamiaceae. For example, in *Lavandula spica* L., *Salvia officinalis* L., *Salvia sclarea* L. and *Nepeta hindostana* (B. Heyne ex Roth) Haines, the development of megagametophytes starts with the four megaspores of the tetrad or only with the sub-micropylar one; or the micropylar and chalazal megaspores; or the sub-micropylar and chalazal megaspores; or the epi-chalazal and chalazal megaspores [41].

We observed that the synergids in the three studied *Saturea* species were hook-shaped, with well-shaped filiform apparatus. This characteristic was also identified in other Lamiaceae species [41] and can also be considered as a typical feature of all species from the Lamiaceae family.

The described characteristics of the reproductive sphere of the studied *Satureja* species define them as sexually reproducing species. This reproductive type ensures stability in the size of the populations of these species. The strict sexuality found in the three target species (no apomixis was observed) limits their adaptive abilities. Combined with the estimated low seed viability, this will affect the reproductive potential of their populations.

The high estimated percentage of fertile pollen in the three studied *Satureja* species is prerequisite to the successful pollination, fertilization and subsequent embryo (seed) formation. It is a result of the observed normal running of the processes of microsporogenesis and development of the male gametophyte. During the microsporogenesis ending predominantly with the formation of tetrahedral microspore tetrads, deviations were not observed. The development of male gametophytes in the pollen was successful, without significant degeneration. Despite this high percentage of viable mature pollen established in the three studied *Satureja* species and sufficient for normal fertilization and embryo (seed) formation, the estimated seed viability was low. A reduction in the percentage of viable seeds has also been found in other Lamiaceae species. For example, for *Salvia nemorosa*, Daskalova [22] estimated that a rather high percentage of seeds were empty and sterile (about 25%). According to the cited author, the large percentage of empty seeds obtained is due to degenerative processes occurring in the female generative sphere, affecting single macrospore tetrads and elements of the embryo sac (most often the egg cell and mature embryo sacs). Our results with the tested *Satureja* seeds showed that the percentage of empty seeds varied from 4.3 to 11.45% (Figure 5), while the percentage of unstained seeds varied between 64.52% and 82.35%. These results indicated that the high percentage of nonviable seeds in *Satureja* is probably due to the impact of environmental conditions rather than to deviations in megagametogenesis and the development of female gametophytes. The high percentage of mature uncolored seeds indicated that the processes leading to the formation of embryos proceeded normally, and the non-viability was a result of secondary disturbances in the quality of the obtained seeds under the influence of environmental factors. Between these factors, the fragmentation of the population of the studied species has a strong impact on their reproductive success. Habitat fragmentation was recognized as a major threat to plant–pollinator interactions [46,47,48,49]. Due to avoidance of small plant populations or isolated fragments by wide-ranging pollinators, reproduction may be reduced and eventually lead to extinction [48,49]. All of these factors affect the reproductive potential of the researched *Satureja* species. The results obtained for seed vitality corresponded with the results of the germination energy and germination tests (Table 1). Empty seeds and low vigor explain the low germination test results. Light, as an environmental factor, also affected seed germination. Members of the *Satureja* genus prefer sunny sites with high light intensity, and that is in accordance with the highest seed germination percentage of both species in variant 1 (natural daylight): 39% for *S. pilosa* and 40% for *S. coerulea*.

### 3.2. Scanning Electron Microscopy (SEM) Analysis

#### 3.2.1. Leaves, Stem, and Calyx Surfaces

The micromorphology of leaf, calyx and stem surfaces of *S. coerulea*, *S. kitaibelii* and *S. pilosa* were analyzed by SEM. The microstructure of leaves, nutlets, calyx, and indumentum of *Satureja* species is an important systematic indicator for their taxonomy [50,51]. This research showed that the three *Satureja* species have many common and different micromorphological features. Overall, the indumenta of *S. coerulea*, *S. kitaibelii* and *S. pilosa* were presented by glandular and non-glandular trichomes, which is not unusual because, for Lamiaceae, this is specific [26,30]. Furthermore, morphology and distribution of trichomes can vary between *Satureja* species [26,30]. Peltate-type trichomes were the most frequently observed in the three species, but they were not located in all plant parts of the *Satureja* species. For example, on the stem of *S. coerulea*, only non-glandular trichomes were observed. This is not accidental because, as indicated in previous research on other *Satureja* species, the hairs show an organ preference [28,30]. The capitate-type trichomes were detected only on the surfaces of *S. pilosa*. The three types of non-glandular trichomes were present on *S. pilosa*, two types on *S. kitaibelii*, and one type on *S. coerulea*. Multicellular and unbranched trichomes were observed along the edge of leaves, while unicellular and uniseriate papillae trichomes were observed on the leaf surfaces (adaxial and abaxial) of *S. kitaibelii* (Figure 7G–L). The non-glandular trichomes on the surfaces of *S. pilosa* were several types: simple, short conical trichomes, unicellular, and multicellular (mostly four-cellular) conical trichomes (Figure 7N–R, Figure 8I–K and Figure 9G–I). All described non-glandular trichomes of *S. pilosa* had specific papillose surfaces that were reported for the first time.

The varying shapes, types and specific characteristics of epidermal cells and stomatal complexes of plants are important taxonomic values, and they are successfully used in taxonomy [52]. In this regard, the three studied *Satureja* species showed different forms of main epidermal cells and stomatal complexes that are related with their taxonomic position. The main epidermal cells of *S. coerulea* are irregular to rectangular in shape, with slightly raised or smooth periclinal walls, while in *S. kitaibelii* and *S. pilosa,* the forms of main cells are isodiametric, and the periclinal walls are striated. The cuticle striation is an important diagnostic indication for the ecological adaptation of species and their taxonomy [53]. Previous investigations of foliar epidermal surfaces of species from Lamiaceae showed that epidermal cells varied from irregular, isodiametric, and rectangular forms, and based on these characters, the authors developed a taxonomic key [50]. The stomatal apparatus is another important characteristic of a species [52]. In the present study’s SEM observations of the three *Satureja* species, amphistomatic leaves were detected, which is in agreement with reports for other *Satureja* species [54]. Previous research on leaves of *S. pilosa* reported hypostomatic leaves [26], which was not confirmed in this study. The stomatal complexes of the studied *Satureja* taxa showed distinctive patterns. The stomata of *S. coerulea* are elliptical and submerged in the epidermis, with distinct cuticles, while the stomata of *S. kitaibelii* are round, small, and diacytic. Furthermore, the cells of the stomatal complex of *S. kitaibelii* have specific striations, and the stomata of *S. pilosa* are oval shapes with a double rim of waxes at the level of the epidermis. Micromorphological analyses of epidermis and stomata of other *Satureja* species have been performed previously [26,28,30,55,56]. The conclusion of all cited authors was that the main epidermal cell and stomatal complex show variation, but diacytic-type stomata are a common feature of most species in the Lamiaceae family. Generally, the epidermis, calyx and stem surfaces of Bulgarian samples of *S. pilosa*, *S. kitaibelii*, and *S. coerulea* were analyzed for the first time. The micromorphological analysis revealed that the surfaces of *S. coerulea* are clearly distinct from those of the other two *Satureja* species.

#### 3.2.2. Nutlet (Seed) Surfaces

The taxonomic importance of the micromorphology of nutlet surfaces of *Satureja* species is well-documented [1]. As known in Lamiaceae, the fruits are schizocarpic. After reaching maturity, the fruit is separated into four nutlets (cocci), and the fruit coat and seed fuse together [57]; the fruit is identified as the seed in Lamiaceae. Scanning electron microscopy (SEM) has been widely used to study the surface features of nutlets in *Satureja* species because the micromorphology of the fruits and ornamentation are much more important in taxonomic studies [58]. SEM analysis and stereomicroscope observation in this study on nutlets of *S. coerulea*, *S. kitaibelii*, and *S. pilosa* showed common but distinguishing features. The common features of the nutlets were their colors, forms, and variations in size. In addition, brown longitudinal lines were observed on the adaxial surfaces of nutlets for the three studied species. These marks were here described for the first time for *S. coerulea*, *S. kitaibelii*, and *S. pilosa*. As was described in the Results section, the tips of the nutlets are triangular and beveled on both sides unevenly. For this reason, determining the shapes of the nutlets depends on how they are arranged in space, so they appear differently (Figure 10 and Figure 11). Most of the previous research on *Satureja* micromorphology grouped the nutlets into two main types and several subgroups, but there is no uniform classification [26,59]. For example, according to Husain et al. [26], in the first group were the nutlets with protuberances and without trichomes and oil glands (*S. montana* group) [26]. In the second group, the authors included nutlets with surfaces that were without protuberances, with papillae, trichomes and sessile oil glands, where they placed *S. pilosa* [26]. In another study, Kaya et al. [59] divided *Satureja* nutlets into two main types: those with more or less smooth surfaces and those with sculptured surfaces, respectively. Furthermore, they subdivided the first type into four subtypes: undulate-reticulate, reticulate, reticulate-protuberculate and papillate-tuberculate, where they put *S. coerulea* [59]. The cited authors [59] found tiny, stalked glandular and eglandular hairs at the nutlets’ apex and on the median edge in *S. cilicica*, *S. coerulea*, *S. icarica*, *S. parnassiaca* subsp. *sipylea*, *S. pilosa*, *S. spinosa*, *S. thymbra*, and *S. wiedemanniana*. In this study, glandular trichomes were not found on nutlets of any of the three studied *Satureja* species. The exocarp surfaces of *S. coerulea* and *S. pilosa* have specific papilloma formations, such as the non-glandular trichomes seen on the end of the nutlets of *S. coerulea*, while they are on all surfaces of *S. pilosa*.

Generally, our study showed different characteristics for the surfaces of the nutlets from *S. coerulea*, *S. kitaibelii*, and *S. pilosa*. Therefore, it was difficult to determine into which group the nutlets we studied fell.

## 4. Materials and Methods

### 4.1. Collection of the Plant Materials

Plant materials of the three *Satureja* species were collected from natural populations as follows: *Satureja pilosa*—locality Selce, Stara planina (42°37′04.3″ N 25°33′02.2″ E; 749 masl); *Satureja coerulea*—locality Markovo, Rhodope mountain (42°02′30.7″ N 24°42′54.1″ E; 638 masl); *Satureja kitaibelii*—locality Kostenkovci, Stara planina (42°57′35.7″ N 25°25′00.1″ E; 604 masl).

### 4.2. Embryological Analyses

The reproductive potential of the three studied *Satureja* species was investigated. The main parameters were estimated as follows: (1) structures and processes in the male and female generative sphere; (2) pollen and nutlet (seed) viability. The flower buds, flowers at different development stages, and nutlets were collected from natural populations of the species. For revealing the peculiarities of structures and development of male and female gametophytes, previously described procedures were followed [44]. According to the methods described by Sundara [60], the fixed material was dehydrated by passing it through increasing ethanol solutions and embedding in paraffin wax. The paraffin-embedded material was cut using a rotary microtome Leica RM2125RT into thicknesses of 8 to 12 μm. Permanent slides were prepared. The description of characteristics of structures in the generative sphere of the studied species was made based on observations using an Olympus Light CX21 microscope (Olympus Corporation, Shinjuku, Tokyo, Japan). The microphotographs were taken with an “Infinity lite” digital camera 1.4 Mpx (Lumenera Corporation, Ottawa, ON, Canada).

#### 4.2.1. Pollen Viability

The quality of the produced mature pollen grains was estimated through pollen viability according to an acetocarmine test [61]. The mature pollen grains in 30 anthers from different individual plants per species were treated with a solution of 1% acetocarmine. On the basis of the intensity of staining from the acetocarmine solution, the pollen grains were classified as follows: (1) viable pollen grains (pollen grains stained in red, with clear reticulated sculpture and strongly distinguished vegetative and generative cells or sperm); (2) nonviable pollen grains (colorless or transparent pollen grains). The fertile and infertile pollen grains were counted on a visible field using a light microscope (described above) at magnification 100× or 400×. The pollen viability was calculated and presented in percentages.

#### 4.2.2. Nutlet (Seed) Viability Testing

Nutlet (embryo) viability was assessed using a tetrazolium test [38]. This topographical tetrazolium method differentiates live from dead seeds based on the activity of the respiration enzymes in seeds. When nutlets are in contact with a solution of 2,3,5-triphenyl tetrazolium chloride, the activity of dehydrogenase enzymes increases. The hydrogen ions reduce the colorless tetrazolium solution into a compound named formazan. As known, formazan stains the cells (respiring) in red if they are alive, and the dead cells remain colorless. Thus, the staining pattern after application of the tetrazolium test reveals the live and dead areas of the embryo and enables one to determine if seeds have the capacity to produce normal seedlings [37]. Approximately 100 mature seeds per species were used, and the procedure for preparation of nutlets was as previously described [45]. According to criteria described by Moore [39], the viable embryos display entire embryo staining (in red or in pink) or staining of their basal part—the root; the nonviable embryos display abnormal or no staining.

#### 4.2.3. Testing of Nutlet (Seed) Germination under Different Light Regimes

The test for germination with different wavelengths (colors) of light was conducted under light intensity of 250 micromoles. A photoperiod of 16 h light and 8 h dark was used. The tested nutlets were arranged in Petri dishes, and they were positioned at a distance of 60 cm from the light sources. The temperature was set at 23 °C during the day and 20 °C at night. Fifty seeds per species were used in each Petri dish in three replications for each variant. Every day, distilled water was added to ensure the seeds remained moist. The experimental design was as previously reported [45]. (1) The first variant (V1) was natural daylight; (2) the second variant (V2) was fluorescent white light with the addition of LED red and blue in a ratio of 7:1; (3) the third variant (V3) was fluorescent white light with the addition of LED red and blue in a ratio of 4:1; and (4) the fourth variant (V4) was fluorescent white light. According to the procedure described in BDS, GE was measured on the fifth day, and total G was measured on the tenth day. All data were analyzed with descriptive statistics, performed using the averages and standard deviation (SD) of the germination energy (%) and total germination (%) obtained from the samples in triplicate. The wavelengths of the light color specifications were reported previously [45].

### 4.3. Scanning Electron Microscopy (SEM) Analysis

For this investigation, an FEI Quanta 600 scanning electron microscope (SEM) at the Microscopy Facility of Oregon State University in the United States was utilized. Sample preparation involved placing small samples into a fixative consisting of 1% paraformaldehyde and 2.5% glutaraldehyde in 0.1 M sodium cacodylate buffer of pH 7.4. The samples were soaked in fixative for 2 h, followed by two rinses in 0.1 M Cacodylate buffer for 15 min each, and then underwent a dehydration series in acetone (10%, 30%, 50%, 70%, 90%, 95%, and 100%), with each step lasting 10–15 min. The samples were then subjected to critical point drying (two "bomb flushes" at chamber pressure to 5 °C, followed by filling the chamber with CO_2_). After venting for 5 min, the procedure was repeated. The dry samples were mounted onto an aluminum SEM stub using double-stick carbon tape and sputter-coated with a Cressington 108A sputter coater from Ted Pella with Au/Pd, 60/40 mix. The shape and surface structure of the nutlets (seeds) of the species were described morphologically. In this study, the terminology and classification described by Barthlott and Ehler [62] were adopted.

## 5. Conclusions

This was the first study of the reproductive capacity and micromorphological (SEM) characteristics of leaf, calyx, stem and nutlet surfaces of three Balkan-endemic *Satureja* species (*S. coerulea*, *S. kitaibelii*, and *S. pilosa)*. This study revealed that all three species reproduce sexually. The established characteristics of the processes in the reproductive sphere suggest low plasticity of the species and is probably the reason for their endemism. Furthermore, the results showed that the reproductive potential of the populations of the three Balkan-endemic *Satureja* species depended on the environmental conditions and the established fragmentation of their habitats. Indeed, the balanced processes and stable structures in the male and female generative spheres, combined with high pollen viability, as observed in this study, provide high reproductive potential, but the produced nutlets had low viability and germination. Furthermore, high numbers of nutlets were empty with undeveloped embryos, which is a major problem for species distribution. The habitats of these endemic species are fragmented, found on dry grasslands, rocky slopes, and stony places where environmental conditions are rather extreme. Therefore, the unfavorable conditions affect pollination and seed production of *S. coerulea*, *S. kitaibelii* and *S. pilosa*. Consequently, the most efficient approach for preserving these species would comprise measures aimed at conserving their habitats and gene pool, such as (1) monitoring the state of the populations; (2) limiting anthropogenic pressure on the populations; (3) establishing in situ and ex situ collections; and (4) developing the three species into field crops for EO production.

Overall, the SEM analyses of *S. coerulea*, *S. kitaibelii*, and *S. pilosa* showed variability between the three species. *Satureja coerulea* microstructural surfaces (leaves, calyx, stem) were clearly dissimilar to those of the two other *Satureja* species. Regarding the nutlet surfaces, this study established that the exocarp surfaces of *S. coerulea* and *S. kitaibelii* had a reticulate convex-type surface, while *S. pilosa* exocarp was a tabular to slightly convex type, without bubble-like cells. The seeds of both *S. coerulea* and *S. pilosa* had specific papilla formations that were not found in *S. kitaibelii*. Furthermore, the non-glandular papillae in *S. coerulea* were observed only on the back side of its nutlets, while those in *S. pilosa* were observed over the entire surface of the nutlets.

## Figures and Tables

**Figure 1 plants-12-02436-f001:**
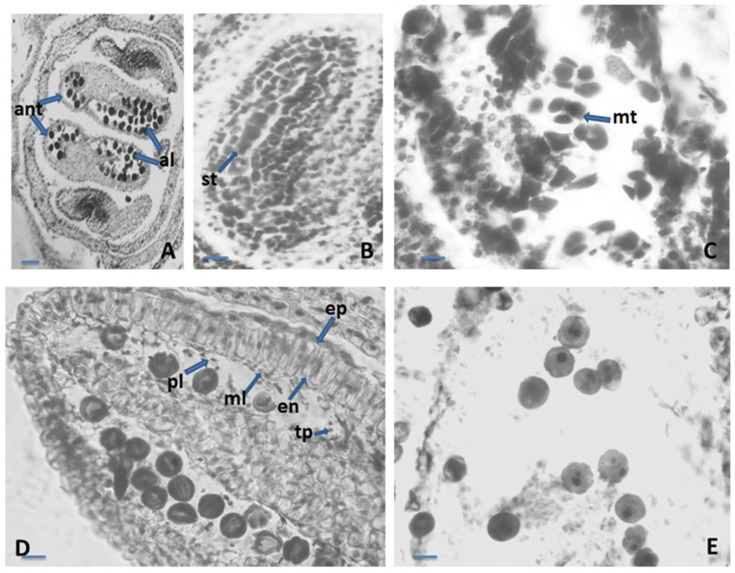
Anther and development of the male gametophyte. (**A**)—tetrasporangiate anther; (**B**)—sporogenous tissue in the anther locule; (**C**)—microspore tetrads in the anther locule; (**D**)—one-nucleate pollen grains and anther wall with epidermis, endothecium, middle layer and tapetum forming placentoids; (**E**)—two-celled mature pollen grains, ant—anther, al—anther locule, st—sporogenous tissue, mt—microspore tetrad, ep—epidermis, en—endothecium, ml—middle layer, tp- tapetum, pl—placentoid. Scale bar: for (**A**) = 50 μm; for (**B**–**E**) = 20 μm.

**Figure 2 plants-12-02436-f002:**
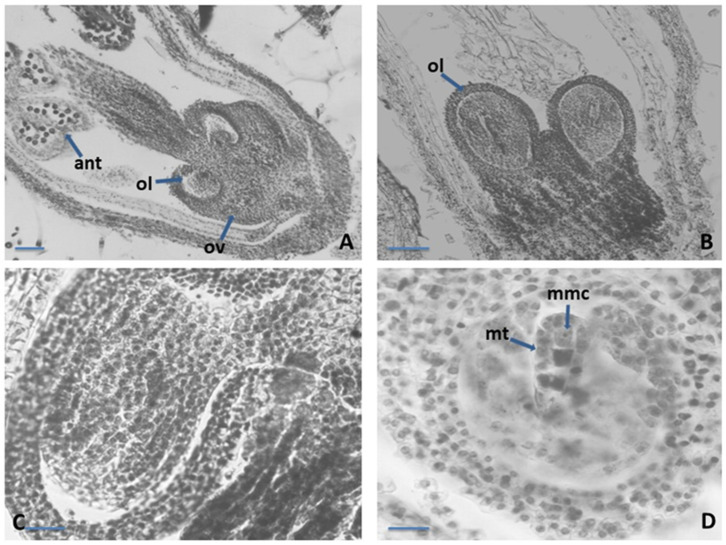
Ovule and development of the female gametophyte. (**A**)—Structure of the *Satureja* flowers; (**B**)–Ovary locules with anatropous unitegmic ovule in each locule; (**C**)—Anatropous unitegmic ovule; (**D**)—Linear macrospore tetrad in the ES cavity with macrospore mother cell; ov—ovary, ant—anther locule, ol—ovary locule, mt—macrospore tetrad, mmc—macrospore mother cell. Scale bar: for (**A**) = 50 μm; for (**B**–**D**) = 20 μm.

**Figure 3 plants-12-02436-f003:**
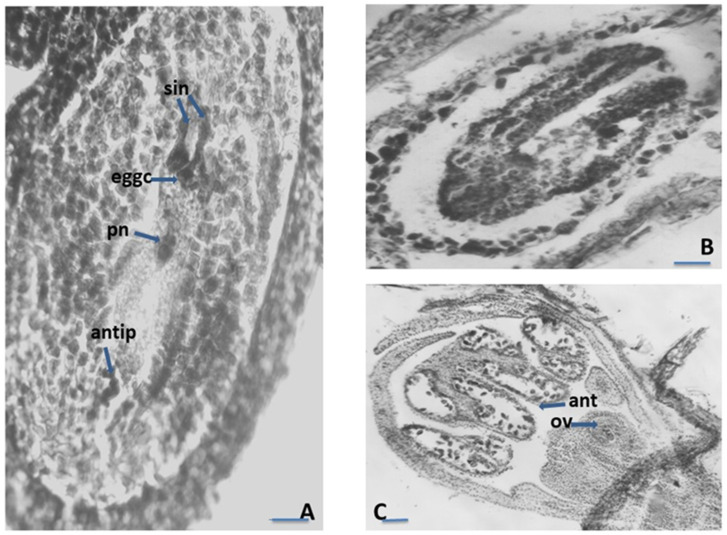
Ovule and development of the female gametophyte. (**A**)—Mature *Polygonum*-type embryo sac (ES); (**B**)—Embryo with cotyledons in ES cavity; (**C**)—Proterandry in the *Satureja* flower; syn—synergids, eggc—egg cell, pn—polar nucleus cell, antip—antipodal, ov—ovule, ant—anther locule; Scale bar: for (**A**,**B**) = 20 μm; for (**C**) = 50 μm.

**Figure 4 plants-12-02436-f004:**
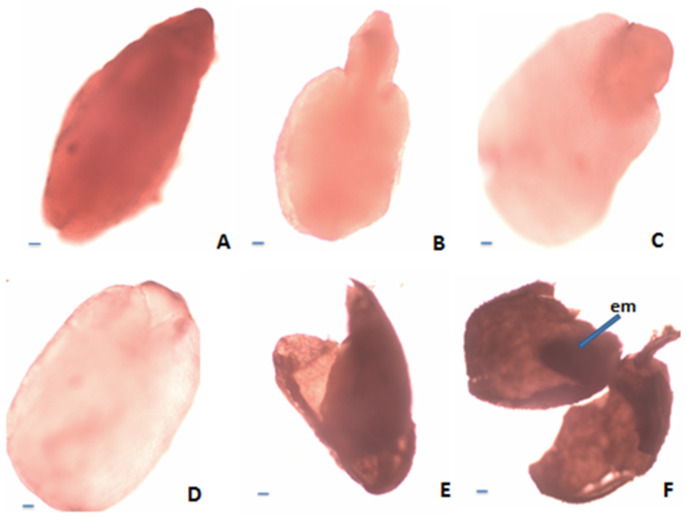
Estimation of nutlet (seed) viability according to tetrazolium test: (**A**)—Viable embryo (stained in dark red); (**B**)—Viable embryo (stained in pink); (**C**)—Viable embryo (only the root stained in red); (**D**)—Nonviable embryo (unstained); (**E**)—Empty seed; (**F**)—Seed with undeveloped embryo. Scale bar = 100 μm.

**Figure 5 plants-12-02436-f005:**
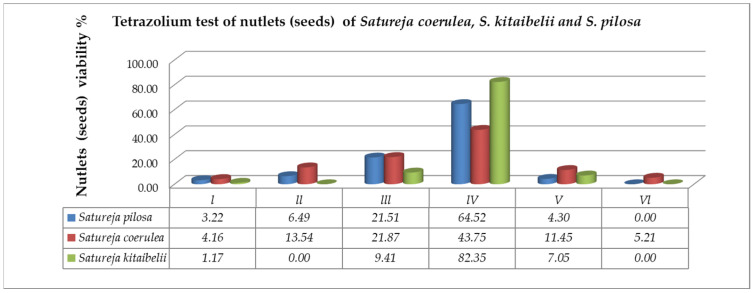
Evaluation of nutlet (seed) viability in classes after TZ testing on the studied *Satureja* species: Class I—seeds with embryos stained in dark red; Class II—seeds with pink colored embryos; Class III—seeds with embryos in which only the root is stained in red; Class IV—seeds with colorless embryos; Class V—empty seeds; Class VI—seeds with undeveloped (dried) embryos.

**Figure 6 plants-12-02436-f006:**
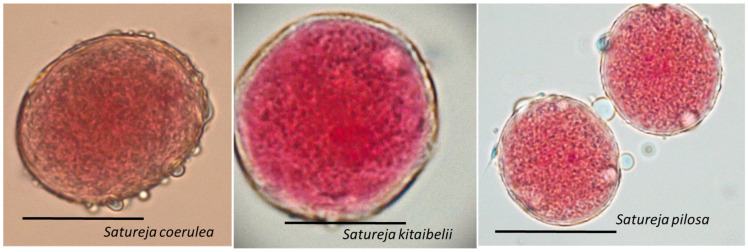
Viable pollen of *Satureja coerulea*, *S. kitaibelii* and *S. pilosa.* The photos were taken with LM Motic DMA (×40); Scale bar = 100 µm.

**Figure 7 plants-12-02436-f007:**
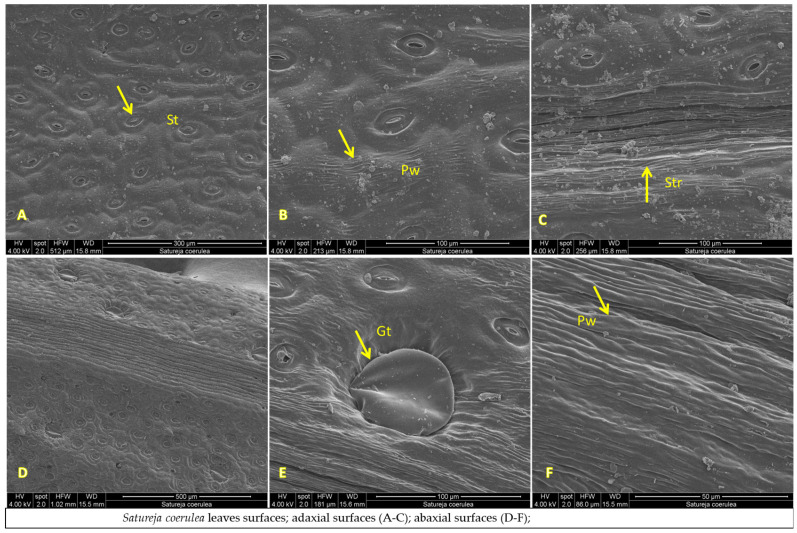
SEM analysis of leaf surfaces of *Satureja coerulea* (**A**–**F**), *S. kitaibelii* (**G**–**L**), *S. pilosa* (**M**–**R**); St—stomata; Gt—glandular trichomes; Ngt—non-glandular trichomes; W—waxes; Pw—periclinal wall; Str—striation.

**Figure 8 plants-12-02436-f008:**
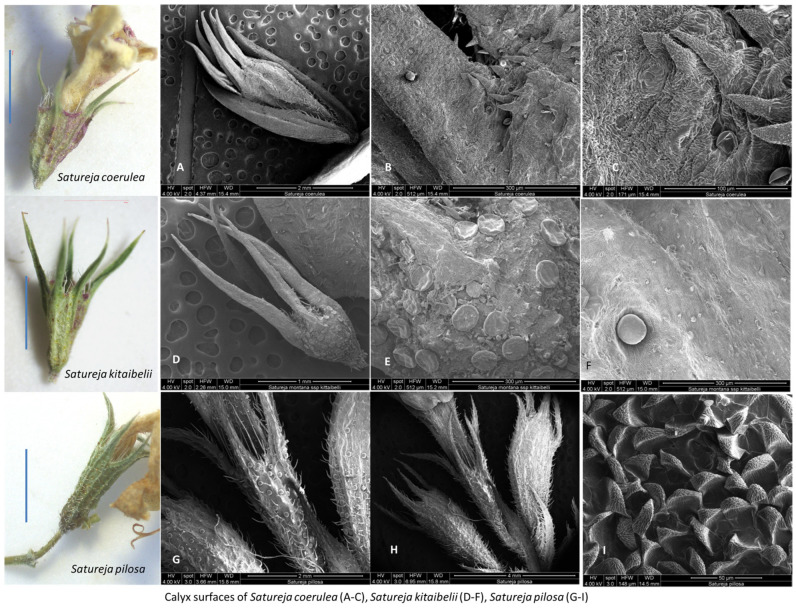
SEM analysis of calyx surfaces of *Satureja coerulea* (**A**–**C**), *S. kitaibelii* (**D**–**F**), *S. pilosa* (**G**–**I**). Scale bar: for the calyx LM = 100 µm. The photos were taken with Stereo Microscope Motic DM 143.

**Figure 9 plants-12-02436-f009:**
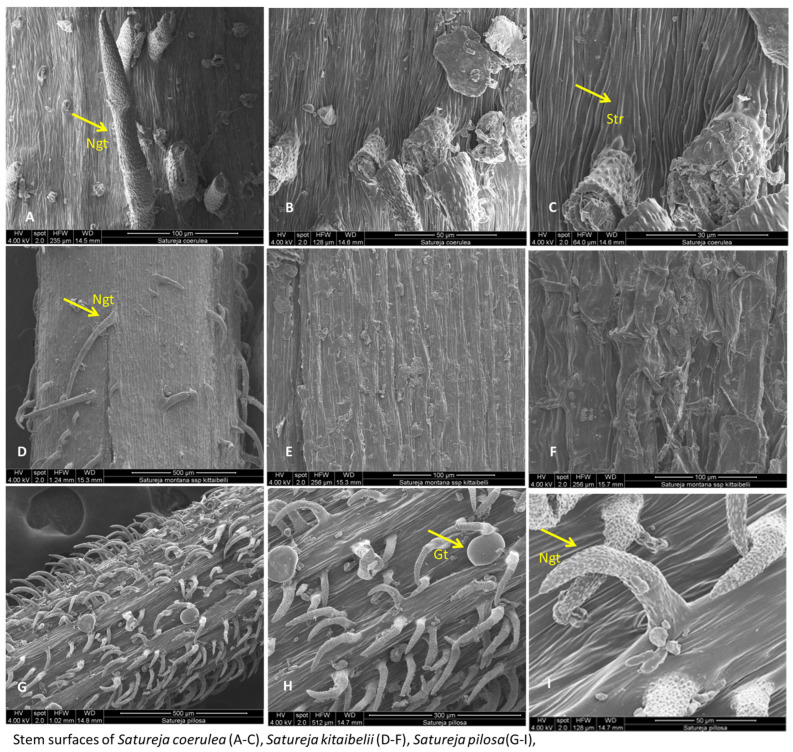
SEM analysis of stem surfaces of *Satureja coerulea* (**A**–**C**), *S. kitaibelii* (**D**–**F**), *S. pilosa* (**G**–**I**). Gt—glandular trichomes; Ngt—non-glandular trichomes; Str- striation.

**Figure 10 plants-12-02436-f010:**
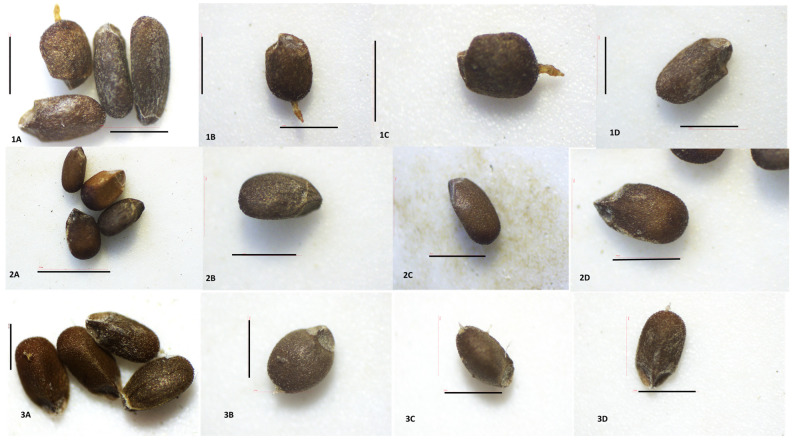
Nutlets (seeds) of *Satureja* species viewed with Stereo Microscope Motic DM 143. *Satureja coerulea* (**1A**–**1D**): (**A**)—view of nutlets from all sides, top, lateral, and tip; (**B**–**D**)—papillae; (**C**)—unevenly triangular tip; (**D**)—adaxial surfaces, papillae; *Satureja kitaibelii* (**2A**–**2D**): A—view of nutlets from all sides top, lateral, and tip; (**B**–**D**)—lateral view, the triangular tip; *Satureja pilosa* 10; (**3A**–**3D**): A—view of nutlets from all sides top, lateral, and tip; (**B**–**D**)—papillae, unevenly triangular tip. Scale bar = 100 µm.

**Figure 11 plants-12-02436-f011:**
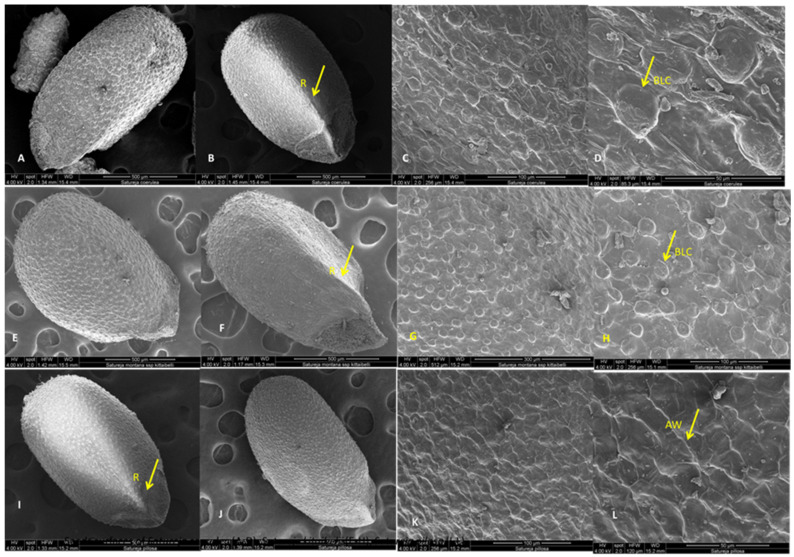
SEM analysis of nutlet surfaces of *Satureja coerulea*, *S. kitaibelii*, and *S. pilosa*; (**A**,**E**,**I**)—the general view; (**B**,**F**,**J**)—triangular edge; (**C**,**D**,**G**,**H**,**K**,**L**)—surfaces. BLC—bubble like cells; R—rib; AW—anticlinal walls.

**Table 1 plants-12-02436-t001:** Germination energy (GE%) and germination (G%) of *Satureja coerulea* and *Satureja pilosa*.

Variants	*Satureja coerulea*		*Satureja pilosa*	
	GE % ± SD	G % ± SD	GE % ± SD	G % ± SD
V1 (N)	24 ± 1.7	40 ± 4.0	21 ± 1.5	39 ± 2.5
V2 (WR7:B1)	20 ± 2.0	30 ± 2.0	9 ± 0.6	18 ± 1.0
V3 (WR4:B1)	31 ± 0.6	33 ± 1.5	13 ± 0.6	26 ± 1.7
V4 (W)	21 ± 1.5	30 ± 2.6	11 ± 1.5	23 ± 3.5

Variant 1—natural daylight—V1 (N), variant 2—fluorescent white light with addition of LED red and blue in a ratio of 7:1—V2 (WR7:B1), variant 3—fluorescent white light with addition of LED red and blue in a ratio of 4:1—V3 (WR4:B1), variant 4—fluorescent white light—V4 (W), SD—standard deviation.

## Data Availability

Data is contained within the article.

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
