# Peer review of "Reproductive Capacity and Scanning Electron Microscopy (SEM) Analyses of the Micromorphological Surfaces of Three Endemic *Satureja* Species from Bulgaria"

_plants, 2023, doi:10.3390/plants12132436_

Round 1
Reviewer 1 Report
This MS presents the results of the first embryological and micromorphological analyses of three Satureja species (S. pilosa, S. kitaibelii, and S. coerulea), aims to establishing the features of male and female reproductive sphere and understanding the mode of reproduction, character, size and state of species populations and delimitation. The MS is well organized, and the data can support the conclusion. Please consider my suggestions to improve the current manuscript.
1. Line 199, add “for” before “S. kitaibelii”. 93.44±3.88 % for S. kitaibelii.
2. Line 206. The legend format of Figure 6 is different from others. Please modify.
3. Line 220. Why not test the Germination energy (GE%) and germination (G%) of Satureja kitaibelii?
4. Line 245. If consistent with the previous text, the font of “Satureja kitaibelii” should be bold.
5. Line 263. The font of “Satureja pilosa” should be bold. And I suggest putting this name on the next page.
6. Line 293. “Satureja” should be represented in italics.
7. Line 349. I think this should be Table 13.
8. A lot of mistakes in the Reference list. For example, reference 8 and 9, use “volume” and “pp”, most other references does not use these two words; reference 10, “DOI: 10.2298/JSC020714106S” should be “https://doi.org/10.2298/JSC020714106S” (also see reference 29); In reference 22, the name of journal “Phytol Balcan.”, should be in italics. In addition, there are also problems with the abbreviations of many magazines. Please check the whole list and modify it.
Author Response
Dear reviewer
Below, please find our “point by point” answers to each of your’s comments and suggestions. We are thankful for constructive comments and suggestions which resulted in an improved manuscript.
Review Report (Reviewer 1
This MS presents the results of the first embryological and micromorphological analyses of three Satureja species (S. pilosa, S. kitaibelii, and S. coerulea), aims to establishing the features of male and female reproductive sphere and understanding the mode of reproduction, character, size and state of species populations and delimitation. The MS is well organized, and the data can support the conclusion. Please consider my suggestions to improve the current manuscript.
- Line 199, add “for” before “ kitaibelii”. 93.44±3.88 % for S. kitaibelii.
Response: Thank you, corrected as suggested.
- Line 206. The legend format of Figure 6 is different from others. Please modify.
Response: Thank you for noticing it. We have corrected them all as suggested.
- Line 220. Why not test the Germination energy (GE%) and germination (G%) of Satureja kitaibelii?
Response: Thank you. We did not test the seeds of S. kitaibelii because most of the collected seeds were empty and therefore, we did not have sufficient number of seeds for 3 replicates. We clarified it in the text 2.1.4.
- Line 245. If consistent with the previous text, the font of “Satureja kitaibelii” should be bold.
Response: Corrected as suggested.
- Line 263. The font of “Satureja pilosa” should be bold. And I suggest putting this name on the next page.
Response: Corrected as suggested.
- Line 293. “Satureja” should be represented in italics.
Response: Thank you, corrected.
- Line 349. I think this should be Table 13.
Response: Thank you. We assume the reviewer meant Figure 13; it was corrected.
- A lot of mistakes in the Reference list. For example, reference 8 and 9, use “volume” and “pp”, most other references does not use these two words; reference 10, “DOI: 10.2298/JSC020714106S” should be “https://doi.org/10.2298/JSC020714106S” (also see reference 29); In reference 22, the name of journal “Phytol Balcan.”, should be in italics. In addition, there are also problems with the abbreviations of many magazines. Please check the whole list and modify it.
Response: Thank you. We corrected all references as suggested.
Reviewer 2 Report
This study presents the results of the embryological and micromorphological analyses of three Satureja species (S. pilosa, S. kitaibelii, and S. coerulea) from the Bulgarian flora. The aim of this study was to establish the features of male and female reproductive sphere, as well as surfaces characteristics of leaves, stem, and calyx in order to understand the mode of reproduction, character, size and state of species populations and delimitation. For the embryological study, flower and flower buds in different developmental stages collected from plants of natural populations and treated according the Classic Paraffin Method.
The manuscript have shown a number of Figures to mark the morphology more clearly. But all the picture must be supplemented with bars, and the picture can be arranged more reasonably and neatly.
Some identical images need to be integrated into one plate;
Cited references are separated by commas.
The reviewer has made some marks in the manuscript.

Author Response
Dear reviewer,
Below, please find our “point by point” answers to each of your’s comments and suggestions. We are thankful for constructive comments and suggestions which resulted in an improved manuscript.
Review Report (Reviewer 2
Quality of English Language
( ) I am not qualified to assess the quality of English in this paper
( ) English very difficult to understand/incomprehensible
( ) Extensive editing of English language required
( ) Moderate editing of English language required
( ) Minor editing of English language required
(x) English language fine. No issues detected
|
Yes |
Can be improved |
Must be improved |
Not applicable |
|
|
Does the introduction provide sufficient background and include all relevant references? |
(x) |
( ) |
( ) |
( ) |
|
Are all the cited references relevant to the research? |
( ) |
(x) |
( ) |
( ) |
|
Is the research design appropriate? |
( ) |
(x) |
( ) |
( ) |
|
Are the methods adequately described? |
(x) |
( ) |
( ) |
( ) |
|
Are the results clearly presented? |
(x) |
( ) |
( ) |
( ) |
|
Are the conclusions supported by the results? |
( ) |
(x) |
( ) |
( ) |
Response: Thank you! We improved the conclusion as follows: This was the first research of reproductive capacity and micromorphologycal (SEM) characteristics of leaf, calyx, stem and nutlets surfaces of Balkans endemic Satureja species (S. coerulea, S. kitaibelii, S. pilosa). This study revealed that all three species are sexually reproducing. The established characteristics of the processes in the reproductive sphere suggest low plasticity of the species, and are probably the reasons for their endemism. Furthermore, the results showed that the reproductive potential of the populations of the three Balkan endemic Satureja species depended on the environmental conditions and the established fragmentation of their habitats. Indeed, the observed in this study balanced processes and stable structures in male and female generative sphere combined with high pollen viability provide a high reproductive potential, but the produced nutlets have low viability and germination. Furthermore, high numbers of nutlets were empty with undeveloped embryo, which is a major problem for species distribution. The habitats of these endemic species are fragmented, found on dry grasslands, rocky slopes, and stony places where environmental conditions are rather extreme. Therefore, the unfavorable conditions affected pollination and seed production of S. coerulea, S. kitaibelii and S. pilosa. Consequently, the most efficient approach for preserving these species would be measures aimed at conserving their habitats and gene pool such as (1) monitoring the state of the populations; (2) limiting anthropogenic pressure on the populations; (3) establishment of in situ and ex situ collections; and (4) development of the three species into field crops for essential oil production.
Overall, the SEM analyses of S. coerulea, S. kitaibelii, and S. pilosa showed variability between the three species. Satureja coerulea microstructural surfaces (leaves, calyx, stem) were clearly dissimilar to those of the two other Satureja species. Regarding the nutlets surface, this study established that the exocarp surfaces of S. coerulea and S. kitaibelii had a reticulate convex type surfaces while S. pilosa exocarp was tabular to slightly convex type, without bubble-like cells. The seeds of both S. coerulea and S. pilosa had specific papillae formations that were not found in S. kitaibelii. Furthermore, the non-glandular papillae in S. coerulea were observed only at the back side of its nutlets, while those in S. pilosa were observed over the entire surface of the nutlets.
This study presents the results of the embryological and micromorphological analyses of three Satureja species (S. pilosa, S. kitaibelii, and S. coerulea) from the Bulgarian flora. The aim of this study was to establish the features of male and female reproductive sphere, as well as surfaces characteristics of leaves, stem, and calyx in order to understand the mode of reproduction, character, size and state of species populations and delimitation. For the embryological study, flower and flower buds in different developmental stages collected from plants of natural populations and treated according the Classic Paraffin Method.
The manuscript have shown a number of Figures to mark the morphology more clearly. But all the picture must be supplemented with bars, and the picture can be arranged more reasonably and neatly.
Response: In agreement with the reviewer's comments, we have added scale bars in the cited figures. In addition, we rearranged and updated all images.
Some identical images need to be integrated into one plate;
Response: In agreement with the reviewer, we updated all images. All figures are arranged by species.
Cited references are separated by commas.
Response: Thank you. We corrected them as suggested.
The reviewer has made some marks in the manuscript.
Response: Corresponding corrections were made in accordance with these comments
Round 2
Reviewer 2 Report
The reviewers believe that Figure 10-12 can be combined into one big picture; I think the authors can improve quality of all Figures.
No http is required in references:
Author Response
Response to Reviewer Comments
Dear Reviewer,
Below, please find our “point by point” answers to Yours 2, second review. Our responses are in bold after each of the comments.
The authors are thankful to the reviewer for the constructive comments and suggestions.
The reviewers believe that Figure 10-12 can be combined into one big picture; I think the authors can improve quality of all Figures.
Response: In an agreement reviewer's comments we have made changes in these figures as follows: Figure 10, Figure 11 and Figure 12 were consolidated in to Figure 10.
No http is required in references:
Response: In according with the Journal requiems, we have added the “http sources” when they are available.